# Periodontal Disease and Senescent Cells: New Players for an Old Oral Health Problem?

**DOI:** 10.3390/ijms21207441

**Published:** 2020-10-09

**Authors:** Ruben Aquino-Martinez, Sundeep Khosla, Joshua N. Farr, David G. Monroe

**Affiliations:** 1Department of Medicine, Division of Endocrinology, Mayo Clinic College of Medicine, Rochester, MN 55905, USA; khosla.sundeep@mayo.edu (S.K.); Farr.Joshua@mayo.edu (J.N.F.); Monroe.David@mayo.edu (D.G.M.); 2Robert and Arlene Kogod Center on Aging, Mayo Clinic, Rochester, MN 55905, USA

**Keywords:** periodontitis, periodontal disease, pathogenesis, inflammation, bacterial infection, immune response, cellular senescence, DNA damage, molecular mechanism

## Abstract

The recent identification of senescent cells in periodontal tissues has the potential to provide new insights into the underlying mechanisms of periodontal disease etiology. DNA damage-driven senescence is perhaps one of the most underappreciated delayed consequences of persistent Gram-negative bacterial infection and inflammation. Although the host immune response rapidly protects against bacterial invasion, oxidative stress generated during inflammation can indirectly deteriorate periodontal tissues through the damage to vital cell macromolecules, including DNA. What happens to those healthy cells that reside in this harmful environment? Emerging evidence indicates that cells that survive irreparable genomic damage undergo cellular senescence, a crucial intermediate mechanism connecting DNA damage and the immune response. In this review, we hypothesize that sustained Gram-negative bacterial challenge, chronic inflammation itself, and the constant renewal of damaged tissues create a permissive environment for the abnormal accumulation of senescent cells. Based on emerging data we propose a model in which the dysfunctional presence of senescent cells may aggravate the initial immune reaction against pathogens. Further understanding of the role of senescent cells in periodontal disease pathogenesis may have clinical implications by providing more sophisticated therapeutic strategies to combat tissue destruction.

## 1. Introduction

Periodontal disease is a bacteria-induced chronic inflammatory condition that gradually deteriorates and destroys the tooth-supporting structures, eventually leading to loss of teeth [1]. About 300–400 bacterial species have been found in samples of human subgingival plaque; however, only 10–20 of these species have a causative role in the pathogenesis of periodontal disease [2,3]. A gradual shift in subgingival microbiota, from predominant Gram-positive aerobes to Gram-negative anaerobic bacteria, has been implicated in the transition from health to disease in periodontal tissues [4]. In response to subgingival bacterial challenge, a local release of proinflammatory signals is produced leading to the activation of the host immune defense system. Although bacteria are essential to initiate inflammation, it is the host immune response against invading pathogens that is the major contributor and ultimate cause of periodontal tissue destruction [5,6]. Interestingly, severe gingivitis lesions can remain stable for long periods, even years or decades, until an “unknown factor” disrupts the host–pathogen homeostasis promoting the progression to periodontitis [6,7,8].

In order to eliminate invading bacteria, neutrophils are recruited into the infected site, where they use oxidative and non-oxidative mechanisms to destroy pathogens [9]. Although neutrophils and released reactive oxygen species (ROS) rapidly protect against invading bacteria, they can also harm healthy host cells leading to oxidative stress-mediated DNA damage over prolonged periods of time [10]. Indeed, it has been recognized that low concentrations of ROS, generated during chronic inflammation, can indirectly cause periodontal tissue destruction through the damage to vital cell macromolecules, including DNA [11]. Along the same line, recent studies have identified that repeated exposure to Lipopolysaccharide (LPS), a component of Gram-negative bacteria membrane, results in DNA damage in different cell types, such as gingival and alveolar bone cells [12,13]. Although DNA lesions can be repaired, those cells exposed to excessive genomic damage undergo either apoptotic cell death or cellular senescence [14,15]. Cells that survive persistent DNA damage acquire a senescent phenotype, which can by itself trigger immune cell recruitment through the dysfunctional upregulation of proinflammatory cytokines. Most senescent cells overexpress Interleukin-6 (IL6), IL1α, IL1β, and IL8, among others, which are collectively known as Senescence-Associated Secretory Phenotype (SASP) [16]. This review focuses on the emerging evidence implicating the potential role of cellular senescence in periodontal tissue deterioration. We emphasize the role of chronic Gram-negative bacterial infection in altering the local periodontal environment. In addition, we set forth a hypothesis that long-term bacterial infection and chronic inflammation facilitates a microenvironment for DNA damage-driven cellular senescence. Finally, we propose a model in which dysfunctional accumulation of senescent cells in periodontal tissues can contribute to exacerbating local immune reaction.

## 2. Cellular Senescence

In 1961, Hayflick and Moorhead identified that the growth of human cells gradually decreased, and eventually arrested, their mitotic capacity when they were cultured for long periods of time in vitro [17]. This limited proliferative lifespan, also known as Hayflick limit, was initially hypothesized as an aging expression at the cellular level [18]. Currently, the original concept of finite proliferation potential is considered a telomere-dependent mechanism, and is associated with the progressive decrease in telomere length as a result of repeated cellular replication (Figure 1).

Given that replicative senescence is a stress-dependent mechanism, cells recognize shortened telomeres as damaged or broken, triggering a DNA damage response (DDR). In this mechanism, ataxia telangiectasia mutated (ATM)-p53 axis activation plays a key function [19]. Although telomere shortening itself does not cause growth arrest, it leads to DNA double-strand breaks that trigger a p53 dependent pathway resulting in p21 activation. In addition to p53/p21 pathway, p16^Ink4a^ is also expressed in senescent cells and has an essential role in establishing and maintaining cell growth arrest [20,21]. Although telomere-dependent replicative senescence is a well-recognized cellular process in vitro, it also occurs in vivo in the context of natural chronological aging. Gradual telomere shortening, a phenomenon that naturally accompanies the process of aging in humans, mice, and other species, is implicated in decreased tissue renewal and age-related tissue decline as a consequence of stem cell exhaustion [22,23]. Although cell division is an essential event that promotes telomere shortening, it is not the only factor implicated in regulating telomere length. Oxidative stress can also contribute to accelerated telomere reduction and replicative growth arrest [24].

### 2.1. Stress-Induced Premature Senescence

Although replicative cellular senescence is strongly associated with excessive shortening of telomeres during aging, normal cells exposed to sub-cytotoxic levels of damaging agents can prematurely undergo cellular senescence. Cigarette smoke, γ-irradiation, chronic exposure to certain Gram-negative bacterial toxins, or moderate concentrations of hydrogen peroxide (H_2_O_2_), can induce many features of replicative senescence in normal cells [25,26,27,28]. Besides proliferative arrest, these and other harmful physical or biochemical stimuli produce enlarged cytoplasmic morphology, apoptosis resistance, and the striking hypersecretion of proinflammatory and proteolytic factors. This senescent-like phenotype induced by unrelated stressors, which is independent of telomere shortening, is called stress-induced premature senescence [29]. Based on the ability of these agents to cause oxidative stress either directly or indirectly, they represent a useful resource to evaluate the impact of stress on cellular senescence. For this reason, many researchers have used various genotoxic agents to accelerate cell senescence. For example, low doses of irradiation (0.5 Gy) can transiently induce DNA damage signaling in human fibroblasts [26]. Of note, such low and transient genotoxic lesions induce a temporal growth arrest, and cells partially recover; however, these cells do not secrete SASP cytokines, such as IL6 [26].

In contrast to transient DNA damage, persistent genomic lesions promote constitutive DNA damage signaling and cellular senescence, which is correlated with increased secretion of inflammatory signals [26,30] In agreement with this observation, several studies have reported that premature senescence can also be induced by exposing human cells to subtoxic H_2_O_2_ concentrations [31,32]. Interestingly, cytolethal distending toxin (CDT), a protein secreted by the periodontal pathogen *Aggregatibacter actinomycetemcomitans* causes a genotoxic effect in different cell types, and also several features observed in cells undergoing replicative or premature senescence [28]. Furthermore, *Porphyromonas gingivalis* LPS induces a detrimental effect on DNA, and accelerated senescence in microglial cells, adipocyte precursors, dental pulp cells, and alveolar bone cells [13,33,34,35]. It should be noted that stress-induced premature senescence is mostly a response to cellular damage; however, cells can undergo the influence of both telomere-dependent and -independent senescence under specific circumstances. One example of this, in which both mechanisms overlap, can be observed in those tissues under constant tissue renewal and exposed to inflammation-mediated oxidative stress. Another example is aging, when oxidative stress induced by low-grade chronic inflammation and age-related replicative senescence can interact and reinforce the senescent phenotype (in part by accelerating telomere shortening) [36,37]. Therefore, cells can undergo accelerated senescence independently of telomere shortening as a result of persistent exposure to DNA damaging stimuli.

### 2.2. Senescence-Associated Secretory Phenotype (SASP)

Although cellular senescence can be induced by repeated mitotic activity and/or several unrelated stressors, a common feature that can occur in most senescent cells is the hypersecretion of multiple signaling molecules, proteolytic enzymes, and other factors. This altered secretome communicates the cellular damage not only to neighboring cells and the surrounding environment, but also alert the innate immune system (see Section 3.1). Senescence-associated factors released into the extracellular space can be classified into soluble signals, insoluble proteins, and matrix-degrading factors, which are collectively known as SASP [16], or senescence-messaging secretome [38]. A complex network of interleukins, chemokines, and other soluble families of growth factors constitute a significant proportion of secreted signaling molecules. One of the most significant and consistently secreted cytokines is IL6 [16]. Although IL8 is also strongly secreted, other cytokines such as IL1α and IL1β are essential modulators of the senescent-associated cytokine network [39].

On the other hand, several proteolytic enzymes associated with both irreversible extracellular matrix degradation and cytokine activation are also produced by senescent cells, including matrix metalloproteinase (MMP)1, MMP3, MMP12, and MMP13 [13,16,40]. Consistently, the upregulation of the recognized senescent biomarker p16^Ink4a^ is associated with the expression of MMP13 in cells isolated from the human degenerated intervertebral disc, and chondrocytes from osteoarthritis lesions [41,42]. The secretion of MMPs by senescent cells can negatively impact and alter their local environment by promoting the proteolytic cleavage of cell membrane receptors, cytokines, and the permanent degradation of extracellular matrix components, such as collagen [16,40]. In addition, the expression of fibronectin, an insoluble matrix glycoprotein that has an important role in cell adhesion, is increased in senescent cells [16]. Besides soluble, insoluble, and proteolytic factors, the generation of high levels of ROS constitutes an integral feature of senescent cells [43].

### 2.3. Paracrine Senescence

In addition to the impact of senescence-associated factors on local inflammation, certain soluble signals can transmit the senescent phenotype to their healthy neighboring cells, a phenomenon known as paracrine senescence or the senescence-induced bystander effect [44,45]. Although the impact on other cells is triggered by the combined effect of multiple soluble signals, specific factors and signaling pathways have a relevant role in this process [46,47]. For example, Acosta et al. demonstrated that cells targeted by senescent factors displayed an increased small mothers against decapentaplegic (SMAD)-2/3 and SMAD1/5 activity. Since SMAD transcription factors are key intracellular targets of transforming growth factor-beta (TGF-β) ligands, this indicates that this pathway plays a relevant role during paracrine senescence [46]. In agreement with this finding, it has been reported that TGF-β inhibition accelerates liver regeneration by inhibiting paracrine senescence [48]. Furthermore, IL6 and insulin-like growth factor binding protein (IGFBP)-4/7 can also produce premature senescence in young cells that contributes to impeded tissue regeneration [49,50]. However, the impact on nearby cells can also be produced independently of cytokines and growth factors. Nelson et al. demonstrated that ROS generation by senescent cells can be transferred between adjacent cells via gap junctions, producing DNA damage in neighboring cells. This genotoxic paracrine effect could explain the formation of scattered clusters of senescent cells observed in hepatocytes [45]. In addition, increased ROS production by senescent cells can also affect nearby cells by activating Nuclear factor-κB (NF-κB) [51]. Therefore, senescent cells are a potent source of proinflammatory cytokines, matrix-degrading factors, and ROS that disrupt their local environment and affect neighboring cells. However, senescent cells and their SASP can have beneficial roles in certain contexts.

### 2.4. Senescent Cells: Friends, Foes, or Both?

From embryonic development to aging, the spatiotemporal presence of senescent cells determines their beneficial or detrimental role. During normal chick embryogenesis, senescent cells have been identified during specific developmental stages, and in particular anatomical structures, such as limbs and pharyngeal arches. Intriguingly, embryonic senescent cells disappear before birth [52]. In this embryological context, senescent cells are eliminated through macrophage-mediated mechanisms followed by tissue remodeling [53]. Of note, human embryos also display these features [54]. After birth and during adult life, it has been proposed that the essential function of senescent cells is to coordinate a multistep process that begins with their own elimination (clearance of damaged cells) and ends with tissue renewal, a sequence called senescence–clearance–regeneration [53]. In agreement with this concept, the regenerative effect is largely mediated by SASP factors in part by promoting stemness in surrounding cells [55]. Ritschka et al. also identified that the positive pro-regenerative effect is produced when factors released by senescent cells acted transiently on treated cells. In contrast, the persistent exposure to those factors produced paracrine senescence resulted in impaired regeneration in vivo.

Consistent with this positive role of senescent cells, Demaria and colleagues demonstrated that senescent fibroblasts and endothelial cells contribute to promoting optimal wound healing through the secretion of specific senescence-associated factors [56]. Although Plasminogen activator inhibitor-1 (PAI1), Chemokine ligand 5 (CCL5), and Chemokine ligand 2 (CCL2) had a moderate expression during early wound healing, Platelet-Derived Growth Factor A (PDGF-A) and Vascular endothelial growth factor (VEGF) were highly secreted by senescent cells at this initial stage. Interestingly, IL6 and TGF-β were not expressed in cells isolated from wounds. A prominent feature of senescent fibroblasts and endothelial cells was their transient presence during wound healing. Therefore, these studies suggest that during adult life the short-term presence of senescent cells can promote an immune-mediated elimination of damaged cells and facilitate tissue regeneration (Table 1).

In contrast to the beneficial effect of senescent cells observed during embryonic development and wound healing, the number of senescent cells intermittently increases during the lifespan. Abnormal accumulation of senescent cells mainly occurs at sites of age-related pathologies and tissues exposed to chronic inflammatory conditions, which is associated with tissue deterioration [16,57]. As a consequence of chronological aging, the burden of senescent cells increases in different tissues in humans, mice, and other species, where they contribute to the development of chronic pathologies including arthritis, osteoporosis, Alzheimer’s disease, atherosclerosis, cancer, and diabetes [58,59] Similar to other age-related pathologies, the etiology of diabetes may be the result of the impact of different aging mechanism, including stem cell exhaustion, chronic low-grade inflammation, macromolecular damage, and cellular senescence. In this scenario, senescent cells can contribute by affecting pancreatic β-cell function, and SASP-mediated tissue damage [60]. Under hyperglycemic conditions, increased β-cell proliferation is produced, which eventually leads to cellular senescence and contributes to insufficient insulin release [61]. Furthermore, Thompson et al. demonstrated that SASP factors have an important role in the immune-mediated senescent β-cell destruction that actively drives Type 1 diabetes in mice [62].

Dysfunctional senescent cells accumulate as a result of defective immune elimination and/or accelerated generation over time [65]. Due to their resistance to apoptosis [66], senescent cells persist in tissues and they can promote a negative impact on their environment through various mechanisms. First, as mentioned above, senescent cells become proinflammatory foci scattered throughout tissues promoting immune cell infiltration that results in collateral tissue damage over time [67,68]. Besides the exacerbation of local inflammation, senescent cells also constitute an important source of factors that contribute to age-related systemic chronic inflammation [69]. Second, besides contributing to aggravated local chronic inflammation, mitotic arrest of resident stem cells associated with repeated cell division during the lifetime, eventually impairs tissue renewal of damaged cells leading to tissue decline [22,23]. Third, the secretion of MMPs by senescent cells contribute to producing detrimental changes in the structure and function of the extracellular matrix observed in age-related pathologies [40]. Fourth, the senescence-associated generation of ROS can promote DNA damage, and subsequently cellular senescence in surrounding cells [45,51]. On the other hand, a reciprocal interaction can be produced between DNA damage and chronic inflammatory conditions. That is, DNA damage can drive senescence-associated chronic inflammation, and chronic inflammatory conditions by themselves facilitate the onset of oxidative stress-mediated DNA damage resulting in a positive feedback loop [70,71]. This concept is supported by the fact that senescent cells are primarily identified in tissues exposed to chronic inflammation, and age-related chronic inflammatory pathologies [16,58,70]. Therefore, the transient presence of senescent cells can promote the regeneration of damaged tissues. In contrast, dysfunctional senescent cell accumulation contributes to exacerbating local inflammation, disrupted extracellular matrix integrity and deficient tissue regenerative potential.

## 3. Cellular Senescence as a DNA Damage Response

Although double-strand breaks are less frequent, they are among the most dangerous and cytotoxic forms of DNA damage [72]. Given that genomic instability, a feature of pre-cancerous cells, is produced when DNA double-strand breaks are not detected or repaired, cells react to genotoxic insults by activating DDR signaling. A key downstream target of DDR is p53 that constitutes a barrier against tumor progression [73]. Transiently arresting proliferation, cells can repair DNA damage maintaining their viability and limiting the transmission of genomic alterations to daughter cells. In contrast, when excessive or irreparable damage is produced, cells can also undergo either programmed cell death or cellular senescence [14,74]. Indeed, cellular senescence and apoptosis are key p53-dependent mechanisms mediating the restriction of tumor development, and evidence indicates that both mechanisms could be suppressed in precancerous tissues [75]. Despite some common intracellular signals are shared by senescence and apoptosis, they represent mutually exclusive processes.

One the one hand, during apoptosis, excessively damaged cells show structural changes in their morphology and appear rounded and smaller in size [76]. In addition, apoptotic cells are rapidly eliminated by macrophages via a process that does not produce an inflammatory reaction [76,77]. By contrast, senescent cells are characterized by their resistance to apoptosis, and display enlarged morphology. It seems that the ultimate fate between apoptotic death or survival in a senescent state depends on the severity and duration of DNA damage [78]. Excessive damage to DNA promotes apoptosis, whereas low and sustained genotoxicity results in cellular senescence [14]. Furthermore, p53 levels, H_2_O_2_ concentrations, and B-cell lymphoma (Bcl)-2 expression are also crucial factors implicated in determining the cell fate decision [10,79]. Another important factor in this fate decision is p21, a cyclin-dependent kinase inhibitor, which can act as an anti-apoptotic factor (inhibitor of apoptosis) [80]. Therefore, cells that survive irreversible DNA acquire a senescent phenotype that protects against malignant transformation.

### 3.1. Immune-Mediated Senescent Cells Clearance

Cells that survive severe and/or irreversible DNA damage confer a high risk for the survival of the organism, because genomic instability predisposes to oncogenesis [81]. Cellular senescence is a potent defense barrier against cancer progression not only because it arrests the proliferation of damaged cells, but also because it triggers an exacerbated inflammatory reaction that activates the immune system [82,83]. Cells that survive irreparable genotoxic stress alert the immune system, attract phagocytic cells and promote their own clearance by triggering a localized proinflammatory cytokine gradient [20,53,65]. In this process, SASP mediators act as cell-intrinsic cues that mediate the recruitment of neutrophils, macrophages, and natural killer cells facilitating the detection and elimination of senescent cells. In addition to cytokines, senescent cells continuously produce High Mobility Group Box 1 (HMGB1) in a p53-dependent manner [84]. Davalos et al. demonstrated that HMGB1 is translocated from the nucleus to the extracellular space, where it also attracts innate immune cells stimulating the removal of senescent cells. Furthermore, they found that HMGB1 regulates the expression of SASP, including IL6, thereby playing an essential role in modulating senescent cells inflammatory activity. This immune-mediated clearance is an important mechanism implicated in restricting the accumulation of senescent cells, which is called senescence surveillance [83,85]. As mentioned above, senescent cells constitute a sustained source of key proinflammatory cytokines, including IL1α, IL1β, IL6, IL8, TNF-α, and monocyte chemoattractant protein (Mcp)-1, among others [16,26,39]. These studies strongly suggest that cellular senescence is a key intermediate mechanism linking local genotoxic damage and immune cell infiltration [86,87]. Therefore, cells that survive irreversible DNA damage promote their own elimination by activating the immune defense system, a process that depending on the context can promote tissue regeneration or degeneration. However, with aging immune-mediated elimination of senescent cells is impaired, which contributes to the accumulation of these dysfunctional cells in old tissues.

## 4. Periodontal Inflammation Creates a Permissive Environment for Cellular Senescence

A hallmark of periodontal disease is chronic inflammation, which is triggered in response to pathogenic bacteria in the subgingival area. Although the inflammatory reaction is initiated to restrict the propagation of microorganisms and protect host tissues, a paradoxical effect is produced. Because the immune-mediated inflammatory response is not specific to kill pathogenic bacteria, this defense mechanism can also damage host periodontal cells. In other words, inflammation does not discriminate between pathogens and healthy resident cells, and consequently, the beneficial effect of repelling infection simultaneously causes undesirable tissue damage. Thus, the emphasis on the etiology of periodontal tissue destruction has been focused on the recognition of invading bacteria and/or their products that trigger an immune reaction. However, in the same context of persistent bacterial infection and chronic inflammation, what happens to those cells that survive such detrimental conditions has been overlooked. In addition, how those cells could deteriorate the periodontal microenvironment is currently an emerging topic. Clear evidence indicates that sustained exposure to oxidative stress produced during inflammation can produce directly or indirectly DNA damage in multiple cell types, including periodontal cells [11,88]. The recent identification of senescent cells in periodontal tissues might provide novel insights into the underlying mechanisms promoting tissue decline [13]. Based on mounting data, there are at least four interconnected factors that could contribute to the accumulation of senescent cells in periodontal tissues. Namely, the persistent Gram-negative bacterial infection, chronic inflammation itself, continuous renewal of damaged tissues, and bacteria-induced local immunosuppression (Figure 2).

### 4.1. DNA Damage-Induced Senescence as a Result of Persistent Gram-Negative Bacterial Infection

DNA damage-driven senescence is probably one of the most underappreciated long-term biological consequences of persistent bacterial infection. The transition from health to disease in periodontal tissues has been associated with a predominantly Gram-negative bacterial infection. Although some of these bacteria can be identified in healthy individuals, *Porphyromonas gingivalis*, *Tannerella forsythia*, and *Treponema denticola*, are strongly associated with chronic periodontal disease in adults [89]. At this point, it is important to emphasize that all of these periodontal pathogens, also called “red complex”, are Gram-negative bacteria identified in subgingival plaque. While it is recognized that pathogens trigger an immunoinflammatory reaction into the infected sites, much less is known about the long-term effects of bacteria and/or their products on DNA damage. Mounting evidence indicates that persistent exposure to LPS, an outer membrane component of Gram-negative bacteria, causes a genotoxic effect in macrophages [90], and blood mononuclear cells [91]. Consistent with this LPS genotoxic effect, a similar response is observed in gingival fibroblasts after the exposure to this Gram-negative membrane component for 48 h in vitro [12]. Remarkably, repeated exposure to *P. gingivalis* LPS causes DNA damage-driven premature senescence in alveolar bone cells, via a process mediated through the p53 activation [13].

Likewise, cytolethal distending toxin (CDT), which is secreted by the Gram-negative periodontal pathogen *A. actinomycetemcomitans*, causes irreversible DNA damage and mitosis inhibition [92,93]. CDT genotoxicity depends on its internalization from the extracellular space into the nucleus of the host cells, where it causes DNA double-strand breaks as a result of its endonuclease activity [94,95]. Consequently, human gingival epithelial cells react to CDT intoxication with DNA damage in the form of double-strand breaks [96]. This permanent genotoxic effect induced by CDT results in either apoptotic cell death or cellular senescence [97,98]. Intriguingly, Fahrer et al. demonstrated that CDT can simulate the effect of relatively low doses of ionizing radiation on DNA damage in human fibroblasts [95]. They also found that CDT not only produced a similar effect as radiation on the expression of several genes, but also that this toxin promoted a higher IL6, Bcl2, and MMP13 expression (in relation to ionizing radiation). Based on their results, the authors suggested that CDT treatment could be a useful model to evaluate the cellular reaction to DNA damage. Therefore, long-term exposure to Gram-negative bacteria and/or their toxins might cause a delayed detrimental effect on periodontal tissues by inducing DNA damage and leading to cellular senescence.

### 4.2. Chronic Inflammation Itself Induces DNA Damage-Driven Cellular Senescence

Chronic periodontal inflammation creates a permissive environment for DNA damage-driven cellular senescence through ROS-mediated oxidative stress. During periodontal bacterial infection, recruited neutrophils secrete many proinflammatory cytokines and release ROS. The former reinforce further recruitment of phagocytic cells, and the latter is one of the most efficient mechanisms to kill invading bacteria. However, besides this bactericidal effect, ROS-mediated oxidative stress generated during chronic inflammation can accelerate cellular senescence through the onset and stabilization of DNA damage [99,100]. Concentration and duration of ROS exposition play a pivotal role in determining cell fate. The biological consequences of oxidative stress can range from cell proliferation to either cellular senescence or apoptotic cell death [99]. These cellular reactions seem to be conserved in different proliferating mammalian cells [101], including mouse and human gingival fibroblasts. Kiyoshima et al. demonstrated that murine gingival fibroblasts exposed to lower H_2_O_2_ concentrations (10 μM) produced increased cell proliferation; however, a gradual decrease in the mitogenic capacity was observed when they were treated with higher concentrations (50 μM) [102]. Along with growth arrest, higher concentrations also induced other senescence-like features such as increased senescence-associated β-galactosidase (SA-β-gal) activity, increased p53 levels, and enlarged cytoplasm. In agreement with these effects, Yu et al. found that H_2_O_2_ not only inhibits proliferation in human gingival fibroblasts, but also induces mitochondrial stress-mediated cell death [103]. Interestingly, Cheng et al. reported that LPS increased the generation of ROS in gingival fibroblasts [12]. As a consequence of higher ROS production, these experiments showed that LPS induced oxidative stress causes DNA damage, which is associated with the expression of both anti-apoptotic, e.g., Bcl2, and other pro-apoptotic proteins. In addition, Xia et al. found that rapamycin decreased the expression of IL6 and IL8 in human gingival fibroblasts infected with *P. gingivalis* by decreasing oxidative stress [104]. These studies suggest that sustained exposure to low H_2_O_2_ concentrations could promote periodontal cell proliferation, whereas intermediate concentrations cause senescence-associated features. In contrast, excessive oxidative stress produced by higher H_2_O_2_ concentrations could cause apoptotic cell death in periodontal cells.

Oxidative stress can indirectly contribute to the pathogenesis of periodontal disease by activating signaling pathways that lead to a “pro-inflammatory state” [11]. It is recognized that excessive ROS generation during inflammation results in periodontal tissue destruction. Indeed, patients with severe periodontitis exhibit a lower antioxidant capacity, which is associated with higher oxidative stress [105]. In addition, increased activity of proinflammatory and oxidative stress markers have been reported in patients with chronic periodontitis compared to healthy controls [106]. Considering the genotoxic effect of ROS, another biological consequence of oxidative stress is genomic instability, which is a hallmark of malignant cell transformation. In agreement with this, bacterial infection has been associated with cancer development by promoting chronic inflammation and generating genotoxic products [107]. These studies raise the possibility that cells that survive long-term exposure to oxidative stress during periodontal inflammation can reach a damage threshold, and eventually activate DNA damage response pathways. Therefore, oxidative stress generated during chronic inflammation can cause cellular senescence by damaging DNA in periodontal cells.

### 4.3. Constant Renewal of Damaged Periodontal Tissues and Replicative Senescence

Senescent cells are not only found in tissues that undergo chronic inflammation, they also reside in tissues with the capacity to be repaired or regenerated [16,82]. Since mammalian cells do not proliferate indefinitely, they undergo permanent mitotic arrest when they reach the end of their proliferative lifespan (as discussed above); gingival epithelial cells similarly undergo replicative senescence after around nine passages in culture [108]. The gingival epithelium acts as a mechanical barrier between bacterial biofilm and gingival tissues, providing a “seal” surrounding the teeth to protect deeper host periodontal tissues against bacterial invasion. When the gingival epithelium is damaged pathogenic microorganisms and/or their toxins gain access into the underlying gingival connective tissue and promote tissue destruction [96]. Besides acting as a physical barrier and secreting antimicrobial agents, the gingival epithelium self-renewal capacity is also a periodontal defense mechanism, as its continuous turnover prevents pathogen invasion [109,110] Thus, alterations in gingival epithelium shedding and renewal could contribute to the development of periodontal tissue destruction [109,110].

Multiple studies have reported that epithelial damage caused by bacterial infection increases cell turnover not only in gingival tissues, but also in non-gingival tissues [111,112]. Indeed, human gingival epithelial cells display accelerated proliferation rates when they are infected with *P. gingivalis* [111]. In contrast, Damek-Poprawa et al. reported that exposure to *A. actinomycetemcomitans* CDT promoted growth arrest in primary human gingival epithelial cells isolated from healthy donors. They also identified that healthy gingival explants exposed to CDT exhibited disruption of the epithelial layers and dissolution of cell junctions. These morphological changes were similar to those observed in gingival explants from patients with periodontitis [113]. Consistent with this, *A. actinomycetemcomitans* CDT topical application promotes mitotic inhibition of rat epithelial cells in vivo [114]. These studies suggest that the self-renewal capacity of gingival epithelium can be impaired directly by certain bacterial toxins, but also indirectly by increasing the proliferating rate eventually leading to accelerated replicative senescence. Therefore, we speculate in this context that constant bacterial challenge may lead to senescence-associated epithelial growth arrest, which might compromise the integrity of the epithelial barrier over time.

Once the gingival epithelial cell barrier has been disrupted, bacteria and/or their virulence factors can reach deeper underlying connective tissues. Gingival fibroblasts are the most abundant cells in gingival connective tissues, and similar to epithelial cells display a limited mitotic capacity. They also have a critical role in sustaining periodontal inflammation [115]. Paez et al. recently reported that primary human fibroblasts from young donors displayed senescence features after extensive cell passaging, including increased p16^Ink4a^ and p21 expression, increased DNA damage, and inhibition of proliferation [116]. Intriguingly, in this study, the expression of inflammatory factors, such as IL6 and IL8, were not identified. These clearly contradictory results are conciliated when we consider that senescence-associated proinflammatory cytokine production is a consequence of persistent DNA damage [26]; p16^Ink4a^ expression as a result of replicative senescence is not associated with the SASP [117]. Supporting this idea, it has been reported that human gingival fibroblasts stimulated with CDT exhibited growth arrest, and increased IL6 protein levels probably as a consequence of DNA damage (as discussed above) [93,118]. In agreement with these studies, Xia et al. demonstrated that rapamycin, an mammalian target of rapamycin (mTOR) pathway inhibitor, not only delayed the onset of senescence features, but also preserved the mitotic potential of healthy human gingival fibroblasts [104]. Taken together, although gingival fibroblasts have a high self-renewal potential, it could be affected during bacteria-induced inflammation limiting gingival regenerative capacity.

In the clinical context, it has been suggested that sustained inflammation during chronic periodontal disease could promote telomere attrition [119]. Indeed, patients with chronic periodontitis displayed shorter leukocyte telomere length, which correlates with disease severity [120]. In contrast, some studies have reported that periodontitis is not associated to telomere shortening [121]. For instance, Sanders et al. found that patients (53–73 years), with and without severe chronic periodontitis, had a similar rate of leukocyte telomere length shortening [122]. These authors suggested that telomere reduction in older individuals could have occurred “earlier in the life course”. These studies suggest that telomere dysfunction might play an important role in periodontal disease progression, but additional factors are required to establish tissue destruction.

### 4.4. Could Bacteria-Mediated Local Immunosuppression Promote Senescent Cell Accumulation?

Impaired immune activity promotes senescent cell accumulation. Besides pathogenic bacteria elimination, the host immune system detects and removes damaged cells, including senescent cells. This process is called senescent cell immune surveillance, in which the release of senescence-associated proinflammatory signals play essential roles in attracting neutrophils, macrophages, lymphocytes and natural killer cells [67,83,85]. However, reduced immunosurveillance of senescent cells accelerates their accumulation, resulting in gradual tissue deterioration and contributing to the development of different chronic pathologies [65]. Despite *P. gingivalis* displays a wide array of invasive mechanisms and different virulence factors, it is important to highlight that its pathogenicity is greatly mediated through the host immunosuppression [123,124,125]. Surprisingly, *A. actinomycetemcomitans* CDT was initially identified as an immunosuppressive factor that contributed to the pathogenesis of periodontal disease [126]. Downregulation of the host immunity by *A. actinomycetemcomitans* could also be mediated through the secretion of Leukotoxin (LtxA), a protein that targets and kills polymorphonuclear cells and monocytes [127]. Interestingly, *Tannerella forsythia* attenuates the early host immune response, which results in its impaired recognition and delayed elimination by the immune defense system [128]. These observations suggest that local host immune suppression caused by bacteria might play an important role in the accumulation of senescent cells by impairing their timely clearance. However, further investigations are required to confirm this hypothesis.

## 5. Can DNA Damage-Driven Senescence Contribute to Increase Local Infiltration of Immune Cells?

Nuclear DNA damage is intrinsically coupled to the release of proinflammatory signals and innate immune activation. In the context of periodontal inflammation, most studies have been focused on the recruitment of immune cells triggered by bacterial recognition [129,130]. Consequently, the emphasis of periodontal tissue destruction has been placed on the immunoinflammatory reaction against bacterial infection. Much less explored has been the role of DNA damage and cellular senescence on the immune defense activation. However, nuclear DNA damage sensors are coupled to a “sterile” inflammatory reaction (non-microbial per se) and the host immune response [131,132]. In fact, DNA damage-driven chronic inflammation is not a new concept in aging and cancer research [133]. Recently, our group identified the presence of increased DNA damage and abnormal accumulation of senescent cells in young periodontal tissues using a murine model [13]. These novel insights suggest that cellular senescence could be an important intermediate mechanism between bacterial infection and innate immune response as senescent cells can by themselves activate the host immune reaction. Therefore, besides being an additional source of proinflammatory signals, senescent cells can potentially aggravate the initial immune reaction against pathogenic bacteria (Figure 3).

### Bacterial Recognition and Nuclear DNA Damage Converge on NF-κB Activation

NF-κB is a crucial transcription factor implicated in the coordination of various signaling pathways that modulate different immune and inflammatory processes. Consequently, NF-κB regulates the cell response to infection, stress, and damage by modulating the expression of proinflammatory cytokines, chemokines, and adhesion molecules [134,135]. Biologically relevant is the fact that different signaling cascades triggered by unrelated stimuli converge at NF-κB. For instance, microbial pathogens and double-strand DNA breaks use distinct signaling transduction pathways to activate NF-κB. In fact, LPS induced NF-κB activation is mainly mediated by Toll-like receptors (TLR), specifically TLR4 [136]; whereas DNA damage signaling is initiated in the nucleus where ATM (ataxia telangiectasia mutated) plays a primary role in the activation of NF-κB [137,138]. These data indicate that unrelated stimuli could reinforce each other, and result in increased inflammatory reaction (Figure 4). Supporting this premise, Gölz et al. identified that hypoxia enhanced the NF-κB activation promoted by *P. gingivalis* LPS in periodontal ligament cells [139]. Another example is the glucose boosting effect on LPS in mononuclear cells. Nareika et al. demonstrated that high glucose produces a strong increment on LPS induced NF-κB activation, which results in higher proinflammatory cytokine and MMPs expression [140]. This is consistent with the ability of high glucose to induce oxidative stress by generating ROS [141]. Of note, NF-κB activation and accumulation coincide with the acquisition of the senescent phenotype, acting as a major regulator of senescence-associated proinflammatory factors including the expression of IL6, IL8, and ICAM1, among others [142,143]. Consequently, NF-κB inhibition decreases oxidative stress DNA damage and delays the detrimental effects of cellular senescence in mice [144]. Therefore, NF-κB activation might be a unifying intracellular event that potentiates the immunoinflammatory reaction to unrelated stimuli, such as nuclear DNA damage itself and bacterial components.

## 6. Clinical Implications and Therapeutic Opportunities

Senotherapy is an encouraging therapeutic approach based on the use of synthetic or natural compounds that selectively eliminate senescent cells, or reduce the expression of SASP factors [145]. Since senescent cells and their factors produce a negative impact on surrounding cells and their environment, an additional beneficial effect of senomorphic drugs could be the restoration of the stem cell regenerative potential. Thereby, this emerging strategy could ameliorate tissue deterioration and promote tissue regeneration in part by inhibiting key anti-apoptotic pathways, or reinforcing the immune system function [145]. Based on the current evidence, potential strategies targeting senescent cells include at least three different approaches [146]. First, inhibition of key transcription factors, such as p53, or signaling pathways controlling the onset and progression of cellular senescence. However, given the role of cellular senescence as an anti-neoplastic mechanism, the downregulation of molecular events leading to senescence can potentially promote oncogenic cell proliferation [147].

Second, an alternative strategy to target specific molecular targets is the elimination of tissue-resident senescent cells. Senolytics are drugs that selectively kill senescent cells by transiently inactivating senescence-associated anti-apoptotic pathways (SCAPs), allowing senescent cells to undergo apoptosis [148]. It has been reported that physical dysfunction and decreased lifespan caused by senescent cells is improved after intermittent senolytic drug administration in old mice [149]. These beneficial effects of senolytics drugs could also be the result of improving stem cell function [22,23]. Third, inhibition of senescence-associated proinflammatory factors by using senomorphic drugs. Senomorphics are agents that interfere with the secretion of SASP factors that promote sterile inflammation, which is produced without inducing apoptosis [150,151]. Senomorphic compounds include mTOR, and NF-κB inhibitors [144,152]. Although senomorphic agents can potentially reduce the secretion of senescence-associated factors, a paradoxical effect could also be produced. That is, since immune-mediated senescent cell elimination is modulated by SASP factors, their inhibition could result in higher accumulation of senescent cells [65].

Since the abnormal accumulation of senescent cells negatively impacts their local environment, and recent evidence indicates that pharmacological agents can modulate SASP activity; these findings are of clinical importance as they can represent an additional approach to traditional periodontal therapy. In fact, some experiments have already demonstrated that pharmacological inhibitors can improve oral and periodontal health. An et al. recently demonstrated that short-term treatment with rapamycin, an mTOR inhibitor, attenuates gingival and alveolar bone inflammation, and also promotes the regeneration of alveolar bone of old mice [153]. Another recent study has demonstrated that mTOR has a central role in regulating LPS induced inflammation in microglial cells [154]. Therefore, senotherapeutic drugs are an emerging and promising approach to delay cellular senescence-related tissue dysfunction, but additional studies are required to better understand their mechanisms of action and efficacy, as well as safety profiles.

In the context of periodontal regeneration, cell-based tissue engineering constructs have been used as alternative strategy of conventional procedures. This regenerative approach involves the expansion of cells in vitro, which are seeded in three-dimensional scaffolds and subsequently implanted into periodontal lesions. Periodontal ligament stem cells (PDLSC) are ideal for periodontal regeneration; however, like other mammalian cells, they undergo senescence. Consistent with this concept, Zheng et al. demonstrated that the proliferative and differentiation capacity of PDLSC also decreases with age [155]. Interestingly, these authors found that conditioned media from young PDLSC improve the proliferative and differentiation capacity of old PDLSC. In a different study, Kuang et al. reported that Metformin, a drug used for the treatment of diabetes, inhibits H_2_O_2_ damaging effects on human PDLSC resulting in decreased oxidative stress-induced senescence [156]. As mentioned above in Section 2.3, TGF-β ligands have a relevant role in cellular senescence. In agreement with this, it has been reported that TGF-β1 induces PDLSC senescence by increasing oxidative stress, and that such effect can be reduced by N-acetyl-l-cysteine (NAC), a ROS inhibitor [157] Of note, Honda et al. found that osteoblastic cells undergo stress-induced premature senescence after implantation in calvarial defects. They also reported that systemic administration of the senolytic D+Q (dasatinib and quercetin) reduced premature senescence in bone defects [158]. These studies suggest that delaying the onset and development of cellular senescence may improve the regenerative potential of periodontal tissues.

## 7. Conclusions

Although bacteria are essential to initiate the inflammatory reaction, it is the host immune reaction that ultimately causes tissue destruction. Emerging evidence indicates that persistent Gram-negative bacterial infection and chronic inflammation itself can facilitate the accumulation of DNA damage on those cells that survive under those conditions. Although genomic lesions can be repaired, cells that survive irreparable genotoxic insults eventually undergo senescence, which is coupled to the hypersecretion of SASP factors and immune defense activation. Since DNA damage-driven senescence can by itself induce an immune response, it is presumed that this novel mechanism can potentially contribute to immune-mediated periodontal tissue destruction. Senotherapy has been developed to selectively kill senescent cells or decrease the deleterious effects of SASP factors. Recently, senescent cells have been identified in the periodontal environment, where they act as an important source of key factors implicated in the etiology of periodontal disease. Another recent study has reported that rapamycin, an FDA-approved drug used as a senomorphic agent, ameliorates inflammation and “rejuvenates” old periodontal tissues. Both studies have established that senescent cells represent potential novel players in the pathogenesis of periodontitis. The selective targeting of these dysfunctional cells and/or their associated factors might represent a novel approach to delay periodontal tissue destruction. Although this is an attractive strategy, further investigation is required to know the potential side effects resulting from senescent cell elimination before this approach can be used in the clinical context.

## Figures and Tables

**Figure 1 ijms-21-07441-f001:**
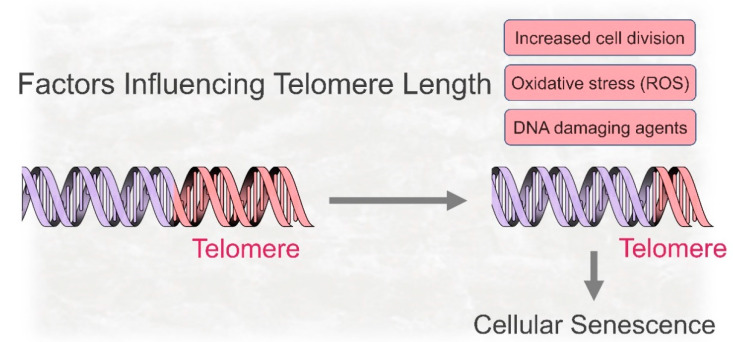
Telomere dependent and independent factors contribute to the onset of cellular senescence. Telomeres gradually shorten as a result of cell division eventually leading to cellular senescence, a phenotype characterized by permanent growth arrest. Although mitotic cell division is the main cause of telomere shortening, this process can be affected or accelerated when cells are exposed to oxidative stress or genotoxic agents. Indeed, unrelated factors, such as radiation, H_2_O_2,_ or certain Gram-negative bacteria products, can induce premature senescence.

**Figure 2 ijms-21-07441-f002:**
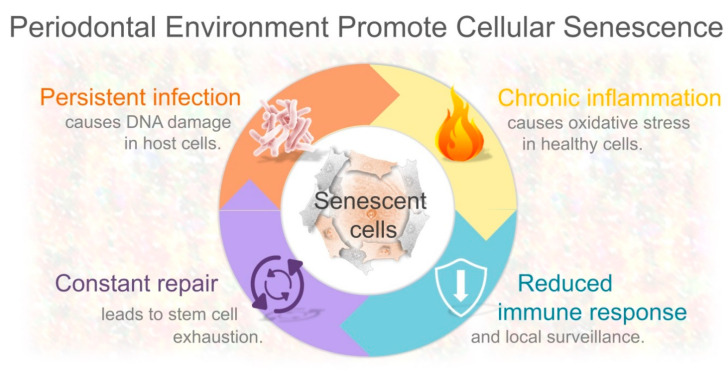
Interconnected factors that may contribute to the accumulation of senescent cells in periodontal microenvironment. Persistent Gram-negative bacterial infection causes chronic inflammation and DNA damage in host cells. Chronic inflammation itself induces oxidative stress in healthy cells. Downregulated local immune response and immune surveillance. Constant renewal of damaged periodontal tissues leads to stem cell exhaustion.

**Figure 3 ijms-21-07441-f003:**
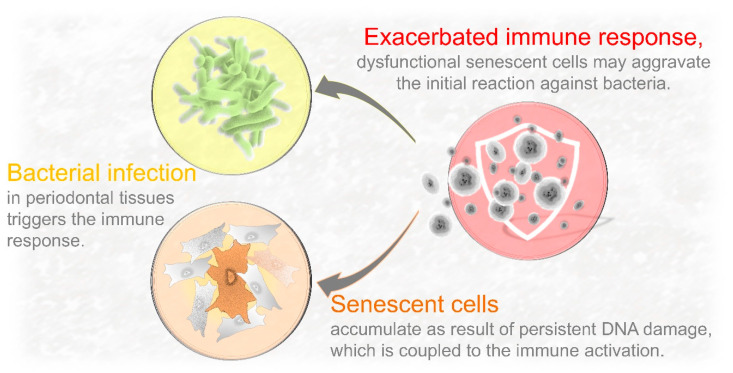
Potential mechanism by which senescent cells aggravate periodontal tissue deterioration. Cells that survive persistent DNA damage acquire a senescent phenotype, which is characterized by the hypersecretion of key proinflammatory cytokines. DNA damage-driven senescence is a crucial intermediate mechanism between DNA damage and host immune activation. Released factors by these dysfunctional cells could aggravate the initial immune reaction against pathogenic bacteria. Thus, gradual accumulation of senescent cells in periodontal tissues during lifetime could contribute to the progression of periodontal tissue deterioration.

**Figure 4 ijms-21-07441-f004:**
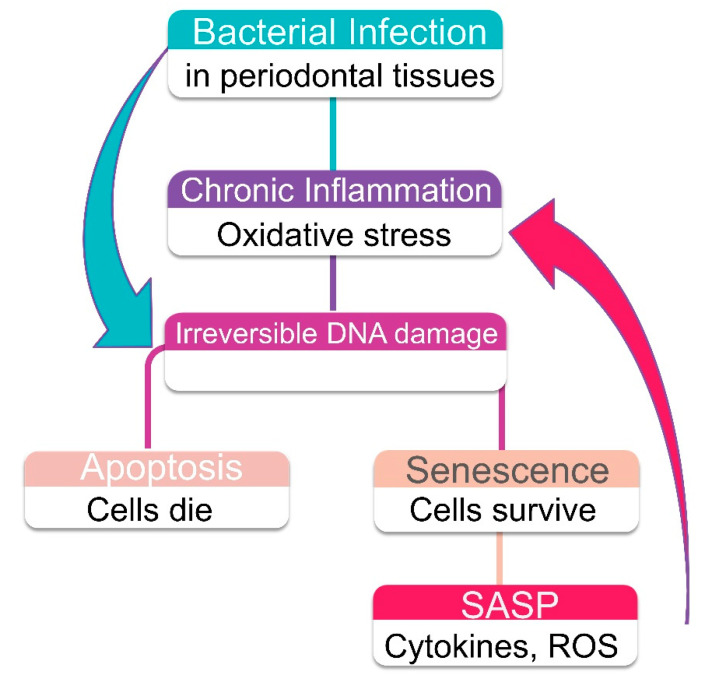
Senescent cells in periodontal tissues are an additional source of proinflammatory factors. Senescence-associated factors can potentially interact and exacerbate bacteria-induced chronic inflammation. NF-κB activation plays an important role in bacterial recognition and nuclear DNA damage signaling.

**Table 1 ijms-21-07441-t001:** The role of senescent cells depends on the context.

Transient Presence and Beneficial Effects	Accumulation and Detrimental Effects
Tumor suppression mechanism [63]	Source inflammatory cytokines (IL-1, TNF-β, IL-6) [16,39]
Embryonic development [52,54]	Source proteolytic enzymes (MMPs) [16,40]
Tissue remodeling and regeneration [53]	Source or ROS [51]
Wound healing [56,64]	Stem cell exhaustion and tissue decline [22,23]

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
