# Peer review of "Periodontal Disease and Senescent Cells: New Players for an Old Oral Health Problem?"

_ijms, 2020, doi:10.3390/ijms21207441_

Round 1

Reviewer 1 Report

In this review, authors focused on the emerging evidence implicating the potential role of cellular senescence in periodontal tissue deterioration. In addition, they set a hypothesis that long-term bacterial infection and chronic inflammation facilitate a microenvironment for DNA damage-driven cellular senescence. Furthermore, authors propose a model in which dysfunctional accumulation of senescent cells in periodontal tissues can contributeto exacerbate local immune reaction. Overall, this review is well written and the topic itself is interesting and could be suitable for the scope of the journal. However, there are a few minor comments that could improve the quality of this manuscript:

  1. The figures are very clear and of decent quality – well done! However, I wonder if they could include an additional figure created by e.g. Adobe Illustrator to better illustrate, in more details, the factors influencing the telomere length as it is central to the focus of this review.
  2. It would be interesting to the readers to discuss the impact of senescent cells on some common periodontal lesion-related systemic diseases, such as diabetes.
  3. I suggest adding a table to summarize both favorable and unfavorable effects of senescent cells that fits to section 2.4. This would make it easier to read.
  4. Recently, it has been reported that TGF‑β1 may serve a crucial role in the senescence of PDL stem cells, which are vital for the regeneration of the periodontal tissues. In the therapeutic opportunities, authors could discuss the potential clinical applications of harnessing stem cell senescence, for instance, in the fabrication of hard tissue for the management of PDL lesions.

Author Response

We would like to thank the Editors and Reviewers of IJMS for their thoughtful comments on our review manuscript. We have carefully considered each of the comments and modified the manuscript along the lines recommended by the Reviewers.

As detailed below in the specific responses to the reviewer’s queries, we have added new figures and text to the manuscript. Our responses to each query are highlighted in yellow below. We again thank the Editors and Reviewers for these comments and we feel this manuscript has been greatly improved as a result of this excellent review process.

REVIEWER 1

1. The figures are very clear and of decent quality – well done! However, I wonder if they could include an additional figure created by e.g. Adobe Illustrator to better illustrate, in more details, the factors influencing the telomere length as it is central to the focus of this review.

Thank you for the insightful comment. We have added a new figure 1, where emphasize that cell replication gradually cause telomere shortening. In addition, we also emphasize that telomere length can also be affected by extrinsic factors. This change was made in lines 70-76.

2. It would be interesting to the readers to discuss the impact of senescent cells on some common periodontal lesion-related systemic diseases, such as diabetes.

Thank you for your interesting comment. We have included a paragraph where we describe how senescent cells may contribute to the development of diabetes. This change was made in lines 203-211.

3. I suggest adding a table to summarize both favorable and unfavorable effects of senescent cells that fits to section 2.4. This would make it easier to read.

We have included a new table 1 that summarize the positive and negative effects of senescent cells. This change was made in the line 196 and line 213.

4. Recently, it has been reported that TGF‑β1 may serve a crucial role in the senescence of PDL stem cells, which are vital for the regeneration of the periodontal tissues. In the therapeutic opportunities, authors could discuss the potential clinical applications of harnessing stem cell senescence, for instance, in the fabrication of hard tissue for the management of PDL lesions.

Thank you for your interesting comment. We agree with reviewer 1. As we had mentioned in the line 158, TGFβ ligands play a crucial role in the development of senescence. Considering that cell-based tissue engineering periodontal tissue regeneration requires cell expansion in vitro, which eventually leads to cellular senescence, we have also included in the new paragraph evidence related to the potential beneficial effects of delaying the onset of senescence. For that reason, we have included a new paragraph the section 6, clinical implications and therapeutic opportunities. This change was made in lines 552-569.

Reviewer 2 Report

General comments:

The authors hypothesize that sustained Gram-negative bacterial challenge, chronic inflammation itself, and the constant renewal of damaged tissues create a permissive environment for abnormal accumulation of senescent cells. In this review, they provide a model to explain the role of senescence on the periodontal disease.

Major comments:

1. Introduction: The authors mentioned that “A gradual shift in subgingival microbiota, from predominant Gram-positive aerobes to Gram-negative anaerobic bacteria, has been implicated in the transition from health to disease in periodontal tissues [4].” But I felt most of the review discuss with the Gram-negative. Please write a sentence to explain that this review mainly focused on Gram-negative.

2. As the authors proposed, the senescence plays an important role in periodontal disease. The treatments or antibacterial agent, clinical drugs that modulate the senescence should be discussed in this review.

Minor comments:

1. Figure 2. The graph is too brief to understand if reader did not read the figure legends. Some concepts or terms did not appear in the graph. Please enhance the graph to cover the description of figure legend.

2. Figure 3. Please add the “Periodontal cells or tissues” to the figure and figure legend. Otherwise, it just look like a common review for senescence to common cells.

Author Response

We would like to thank the Editors and Reviewers of IJMS for their thoughtful comments on our review manuscript. We have carefully considered each of the comments and modified the manuscript along the lines recommended by the Reviewers.

As detailed below in the specific responses to the reviewer’s queries, we have added new figures and text to the manuscript. Our responses to each query are highlighted in yellow below. We again thank the senior Editors and Reviewers for these comments and we feel this manuscript has been greatly improved as a result of this excellent review process.

REVIEWER 2

1. Introduction: The authors mentioned that “A gradual shift in subgingival microbiota, from predominant Gram-positive aerobes to Gram-negative anaerobic bacteria, has been implicated in the transition from health to disease in periodontal tissues [4].” But I felt most of the review discuss with the Gram-negative. Please write a sentence to explain that this review mainly focused on Gram-negative.

Thank you for the interesting comment. We have included this new sentence in lines 57-59.

2. As the authors proposed, the senescence plays an important role in periodontal disease. The treatments or antibacterial agent, clinical drugs that modulate the senescence should be discussed in this review.

We agree with the Reviewer. Although the role of cellular senescence in periodontal tissue destruction is an emerging topic, and for that reason only few papers have been published, we have included some of the most relevant approaches described to combat cellular senescence in periodontal tissues/cells in the current literature. Among these strategies are the use of Rapamycin and Metformin, both currently used as senolytic drugs. Because this comment is closely related to the point 4 of the other reviewer, we included this information in new paragraph in section 6, clinical implications and therapeutic opportunities. This change was made in lines 552-569.

3. Figure 2. The graph is too brief to understand if reader did not read the figure legends. Some concepts or terms did not appear in the graph. Please enhance the graph to cover the description of figure legend.

We agree with reviewer 2. We have included in the new Figure 3 relevant concepts to understand the potential role of senescent cells in aggravating the initial immune response against bacterial infection. This change was made in the line 479.

4. Figure 3. Please add the “Periodontal cells or tissues” to the figure and figure legend. Otherwise, it just look like a common review for senescence to common cells.

Thanks for your comment. We have included in the new figure 4 “Bacterial infection in periodontal tissues” and we also added “in periodontal tissues” in the figure legend. This change was made in lines 510-512.